# Morphological Structure and Physiological and Biochemical Responses to Drought Stress of *Iris japonica*

**DOI:** 10.3390/plants12213729

**Published:** 2023-10-30

**Authors:** Xiaofang Yu, Yujia Liu, Panpan Cao, Xiaoxuan Zeng, Bin Xu, Fuwen Luo, Xuan Yang, Xiantong Wang, Xiaoyu Wang, Xue Xiao, Lijuan Yang, Ting Lei

**Affiliations:** 1College of Landscape Architecture, Sichuan Agricultural University, Chengdu 611130, China; 13438330788@163.com (Y.L.); caopanpan0905@163.com (P.C.); 13551113479@163.com (X.Z.); 18215619872@163.com (B.X.); a750632558@163.com (F.L.); 13228125187@163.com (X.Y.); wxt1231@stu.sicau.edu.cn (X.W.); therealxiaoyuwang@163.com (X.W.); 71044@sicau.edu.cn (L.Y.); ting_lei85@sicau.edu.cn (T.L.); 2Triticeae Research Institute, Sichuan Agricultural University, Chengdu 611130, China; demi821214@126.com

**Keywords:** biochemical indicators, cellular homeostasis, anatomical structure, physiological properties, antioxidant system

## Abstract

Drought is among the most important abiotic stresses on plants, so research on the physiological regulation mechanisms of plants under drought stress can critically increase the economic and ecological value of plants in arid regions. In this study, the effects of drought stress on the growth status and biochemical indicators of *Iris japonica* were explored. Under drought stress, the root system, leaves, rhizomes, and terrestrial stems of plants were sequentially affected; the root system was sparse and slender; and the leaves lost their luster and gradually wilted. Among the physiological changes, the increase in the proline and soluble protein content of *Iris japonica* enhanced the cellular osmotic pressure and reduced the water loss. In anatomical structures, *I. japonica* chloroplasts were deformed after drought treatment, whereas the anatomical structures of roots did not substantially change. Plant antioxidant systems play an important role in maintaining cellular homeostasis; but, as drought stress intensified, the soluble sugar content of terrestrial stems was reduced by 55%, and the ascorbate peroxidase, glutathione reductase, and monodehydroascorbate reductase (MDHAR) activities of leaves and the MDHAR activity of roots were reduced by 29%, 40%, 22%, and 77%, respectively. Overall, *I. japonica* was resistant to 63 days of severe drought stress and resisted drought through various physiological responses. These findings provide a basis for the application of *I. japonica* in water-scarce areas.

## 1. Introduction

In China, 51.4% of the total land area is facing drought, and, even in areas not historically affected by drought, there are frequently uneven precipitation and seasonal droughts [1]. In arid areas, ornamental plants are mostly ground cover and foliage plants, with a relatively homogeneous landscape effect and poor urban landscape quality, so it is important to study multiple species of drought-tolerant ornamental plants. Abiotic stresses, including drought, were reported to have serious effects on plant growth and development [2]. Drought stress responses in plants are characterized by a reduction in soil moisture that limits plant water uptake, while transpiration causes water to evaporate from plants, ultimately leading to death. The earliest leaf physiological responses to drought are the progressive partial closure of stomata, which directly restrains the leaf–atmosphere gas exchange and decreases the ratio of CO_2_ to O_2_ [3]. On the other hand, plants have evolved various mechanisms to alleviate water deficit in drought conditions, including mechanisms at the morphological, physiological, biochemical, and molecular levels [4,5].

Plant structural changes, physio-biochemical responses, and plant drought-adaptation mechanisms under drought conditions are a recent research hotspot. For example, it was recently reported that when rice is exposed to drought stress before or during tillering, the number of tillers and panicles is reduced, and this can even favor genotypic variation [6]. Kebbas et al. [7] investigated the effect of drought on *Acacia* by measuring the growth of plants under drought conditions (leaf number, leaf dry weight, stem length, and leaf area), as well as a series of physiological indicators such as leaf water content, photosynthetic pigments, gas exchange parameters, and leaf nutrients. Jiang et al. [8] explored the biomass allocation characteristics and root morphology responses of *Carex breviculmis* to soil drought at the physiological level. Wang et al. [9] found that when tall fescue was subjected to water stress, the malondialdehyde content, proline content, soluble protein content, soluble sugar content, and peroxidase (POD) activity of plants were higher than those of the control. Based on the above studies, it is clear that the physiological response and adaptation mechanisms of plants to drought stress are complex and systematic. Moreover, the phenotypic variation and biochemical responses among different plants substantially differ.

As a perennial herb, *Iris japonica* is widely distributed in southern China and Japan [10]. It readily forms a single dominant population through seed and vegetative rhizome propagation and has high ornamental and medicinal value [11]. Under drought conditions, the viability of cells within each plant is compromised, and the fact that *I. japonica* is able to tolerate more severe droughts under these environmental conditions makes it potentially economically viable in water-scarce areas [12]. However, to date, studies on the phenotypic and biochemical responses of *I. japonica* under drought stress remain insufficient. Therefore, the aim of this study is to investigate the effects of drought on the morphological, physiological, and biochemical characteristics of *I. japonica*, to provide a scientific basis for *I. japonica* cultivation. In addition, the research results lay the foundation for an in-depth investigation of the drought-resistance mechanism of *I. japonica*. At the same time, it provides an empirical basis and practical guidance for the screening and effective utilization of ornamental plants in arid areas, which is of great significance for local urban greenery planting and management.

## 2. Material and Methods

### 2.1. Plant Materials

The plant material was *I. japonica* obtained from Wenjiang Xiyang Garden Company (Chengdu, China), specifically, three-year-old seedlings. Each pot contained five plants, with 72 pots in total (upper diameter, 15 cm; lower diameter, 13.8 cm; height, 15.7 cm), and the soil used in this study was a sandy clay loam that contained all the necessary nutrients for plant growth. The basic environmental conditions for the experimental plants included a 16 h light (25 °C)/8 h dark (22 °C) cycle, 60% relative humidity, and illumination intensity of 500 μmol m^−2^ s^−1^ photons. The plants were pre-cultured for three months to stabilize the plant roots and growth.

### 2.2. Experimental Design

#### 2.2.1. Processing of Materials

The abovementioned cultivation materials were randomly divided into two groups of 36 pots each, and the plants were placed in the greenhouse of Sichuan Agricultural University for experiments. During the period in which the experiment was conducted, the growth conditions (except for moisture) were the same as during the initial cultivation period. For the control group, 500 mL of water was applied every 3 days, and the excess water naturally flowed out from the bottom of the basin; for the treatment group, there was no watering. A total of four replicate groups were set up for each treatment group of the experiment. The control groups were CK1, CK2, CK3, and CK4, each with nine pots, and the same number of pots was used for the treatment groups.

#### 2.2.2. Collection of Materials

The drought lasted for 63 days (on day 63, the plants in the treatment group were collapsed with yellowed and wilted leaves). According to the number of drought days, the drought was divided into four stages: the early stage (0–9 d), the middle stage (18–27 d), the late stage (36–45 d), and the last stage (54–63 d). The indicators were determined as follows.

(1)For morphological indicators, the plant growth status was observed, including measurements of the following traits: leaf length, leaf width, leaf weight ratio, aboveground stem weight ratio, rhizobia weight ratio, root to weight ratio, and root to shoot ratio after 63 d of drought. The determination of the anatomical structure of the leaves, aboveground stems, and roots was also conducted.(2)For physiological indicators, every 9 days, soil samples were collected to determine soil water content. Plant leaves were collected for determination of leaf cell membrane permeability and proline, soluble protein, reducing sugar, ascorbic acid (AsA), and glutathione (GSH) contents. Whole plant material was collected for determination of tissue water content. Leaves, shoots, and roots of plants were collected for determination of soluble sugar content. Roots of plants were collected for determination of root activity. After 63 d of drought, the leaves of the plants were collected for the determination of photosynthetic pigment, hydrogen peroxide, and malondialdehyde (MDA) contents and peroxidase (POD), superoxide dismutase (SOD), glutathione reductase (GR), ascorbate peroxidase (APX), monodehydroascorbate reductase (MDHAR), and ascorbic acid oxidase (AAO) activities.

### 2.3. Assay Methods

#### 2.3.1. Determination of Morphological Indicators

A combination of straightedge measurement and CAD precision measurement methods was used for the determination of plant parameters, such as leaf length and leaf width.

Fresh weights of leaves, young stems, rhizomes, and roots were measured with a METTLER TOLEDO ME204 scale (Greifensee, Switzerland), with an accuracy of one part in 10,000. The leaf weight ratio, aboveground stem weight ratio, root weight ratio, root weight ratio, and root crown ratio were calculated according to the following formulas: leaf weight ratio = leaf weight/total weight; aboveground stem weight ratio = aboveground stem weight/total weight; rhizosphere weight ratio = rhizomes weight/total weight; root weight ratio = root weight/total weight; root to shoot ratio = (root stalk weight + root weight)/(leaf weight + aboveground stem weight).

For anatomical structure determination, FAA fixative solution (5 mL of formaldehyde, 5 mL of glacial acetic acid, and 90 mL of 50% ethanol) was used to fix leaves, ground stems, and roots by immersing them in the fixing solution, after which samples were sent to Wuhan Sevier Biotechnology Co., Ltd. (DROIDE, Shanghai, China) to be embedded in paraffin wax. Sections were stained with saffron-fixed green, and slices were cut, observed in cross-section with a microscope, and photographed using OLYMPUS cellSens Standard software (V4.1.1).

#### 2.3.2. Determination of Plant Tissue Water Content and Soil Water Content

To determine the water content of leaves, aboveground stems, rhizomes, and roots, the fresh weight (W1) was first obtained, and the dry weight (W2) was obtained after drying samples at 105 °C for 15 min and 80 °C for 48 h. The dry samples were also used for subsequent sugar content determination. Water content was calculated as follows: water content = (W1 − W2)/W1 × 100%.

After collecting the soil, the initial soil weight was recorded (w1), and the soil was weighed again after drying at 80 °C to a constant weight to obtain the weight of the dry soil (w2). The soil water content was calculated as follows: soil water content = (w1 − w2)/w2 × 100%.

#### 2.3.3. Determination of Proline Content

The acidic ninhydrin method was used for the determination of proline content [13]. Proline was extracted with alcohol, which was then heated with an acidic ninhydrin solution (2.5 g of ninhydrin, 60 mL of ice-cold acetic acid, and 40 mL containing 2 mol L^−1^ phosphate that was heated and dissolved at 70 °C, cooled, and stored in a brown reagent bottle) in a boiling water bath for 15 min to produce stable red products. Then, a spectrophotometer was used to obtain the optical density of the test samples at 520 nm. Finally, the content of proline was calculated from the standard curve.

#### 2.3.4. Determination of Cell Membrane Permeability

The conductivity method was used to measure the sample cell membrane permeability using an electronic conductivity meter (FE30, Mettler Toledo) [14]. Ultra-pure water was used as a blank control, and the cell membrane permeability data were obtained by separately measuring each treated sample leaf. Collected *I. japonica* leaves were rinsed in water, dried, and cut into small 0.5 × 0.5 cm pieces. The leaves were divided into eight test tubes, four tubes each for the CK treatment and the drought treatment. The eight test tubes were dried under vacuum for 10 min, subsequently removed, and then kept at room temperature for 30 min, and the exudate conductivity values were determined by an electronic conductivity meter (where the CK measurement was recorded as C1, and the drought group was recorded as T1). Afterward, the tubes were placed in a boiling water bath for 5 min and then cooled to room temperature, and the exudate conductance values were determined (of which the CK measurement was recorded as C1′ and the drought group as T1′) using the following formulas:Leaf membrane permeability of the control group (C) = C1/C1′;
Leaf membrane permeability of the treatment group (T) = T1/T1′.

#### 2.3.5. Determination of Soluble Protein, Hydrogen Peroxide, and Antioxidants

The determination of soluble protein was performed by the Komas Brilliant Blue G-250 staining method; the solution to be tested was reacted with quantitative Komas Brilliant Blue solution, and the absorbance value of the conjugate was measured at 595 nm, allowing the calculation of the soluble protein content [14]. The hydrogen peroxide content was determined using a kit from the Nanjing Jiancheng Institute of Biological Engineering Co., Ltd. (Solarbio, Nanjing, China). The reactants were obtained according to the steps indicated by the instructions, and the absorbance values were measured at 405 nm, enabling the calculation of the hydrogen peroxide content. The MDA content was determined by the thiobarbituric acid method [15].

#### 2.3.6. Determination of Antioxidant Enzymes and Antioxidants

Extracts for antioxidant enzymes and antioxidants were prepared following the methods described by Wu et al., with some modifications [16]. Leaf samples (0.3 g) were ground into a homogenate on ice with 10 mL of precooled phosphate buffer (50 mM; pH 7.8), which was centrifuged at 5000 r/min for 10 min at 4 °C, and the supernatant was collected and used for testing. The nitroblue tetrazolium (NBT) photoreduction method was used, and the absorbance values of the reduced system were measured at 560 nm, enabling the calculation of SOD content as described by Liu et al. [14].

The following enzyme solutions were prepared using the method described by Tan et al., with some modifications [17], and the enzymes were measured to have 1 unit of enzyme activity per minute change in absorbance value of 0.1. Leaf samples (1 g) were ground into a homogenate on ice with 5 mL of pre-cooled phosphate buffer (containing 0.1 mM EDTA, 2% PVP, and 0.3% Triton X-100; pH 7.5), which was centrifuged at 5000 r/min for 10 min at 4 °C, and the supernatant was collected for the assay. For determination of APX activity, the reaction system was 3 mL containing 0.2 mL of supernatant, 1 mM of hydrogen peroxide, 0.5 mM of AsA, and 50 mM of phosphate buffer (pH 7.5). Distilled water was used as a blank, and the change in absorbance at a wavelength of 290 nm was recorded. The reading was recorded every 10 s for 1 min. For determination of GR activity, the reaction system was 2 mL, containing 0.2 mL of supernatant, 1.6 mL of 25 mM phosphate buffer (containing 0.1 mM EDTA; pH 7.8), 0.1 mL of 2.4 mM NADPH, and 0.1 mL of 10 mM GSSG. Distilled water was used as a blank, and the change in absorbance at a wavelength of 340 nm was recorded. The reading was recorded every 30 s for 4 min. For determination of MDHAR activity, the reaction system was 3 mL, containing 0.2 mL of enzyme solution, 0.3 U of AAO, 0.1 mM of NADH, 0.25 mM of AsA, and 50 mM of phosphate buffer (pH 7.5). Distilled water was used as a blank, and the change in absorbance at a wavelength of 340 nm was recorded. The reading was recorded every 30 s for 4 min.

Leaf samples (0.5 g) were combined with 5 mL of 5% TCA, ground into a homogenate on ice, and centrifuged at 5000 r/min for 10 min at 4 °C, and the supernatant was collected for the assay [9,18]. For AsA determination, the method of Arrigoni et al. was used [19]. GSH was determined by the method of Griffith [20].

#### 2.3.7. Determination of AAO Activity

The enzyme solution used for determination of AAO activity was the same as the enzyme solution described in Section 2.3.6. The enzyme solution was 3 mL, containing 0.2 mL of enzyme solution, 0.5 mM of AsA, and 50 mM of phosphate buffer (pH 7.5). Distilled water was used as a blank to record the change in absorbance at 245 nm. Readings were recorded every 30 s for 3 min. Similarly, a change in absorbance per minute of 0.1 was recorded as one unit of enzyme activity.

#### 2.3.8. Determination of Photosynthetic Pigments

For determination of photosynthetic pigments, 0.13 g leaf samples were cut into small pieces in the dark, added to 25 mL of 95% ethanol, and soaked for 48 h in the dark. The absorbance was measured at 665 nm, 649 nm, and 470 nm with 95% ethanol as a blank, and the concentration of each photosynthetic pigment was calculated as follows: content of each pigment (mg·g^−1^) = (*C* × *V*_t_)/(FW × 1000). Here, *C* is the chlorophyll concentration (mg L^−1^), *V*_t_ is the total volume (mL) of the extract, and FW is the fresh weight (g) of the leaves.

#### 2.3.9. Determination of Root Vigor

To assay root vigor, the TTC method was used [13]. The 0.5 g root tip samples from the two treatment groups were processed according to the experimental procedure. Then, each optical density was measured at 485 nm, and the standard curve was plotted. The absorbance at 485 nm was measured following the steps of the TTC test method for the samples. The amount of tetrazolium reduction of the samples was calculated by the formula to derive the root vigor data.

#### 2.3.10. Determination of Soluble Sugars and Reducing Sugars

Preparation of extract followed the method of Zhang et al. [13]. The leaves of the remaining plants were all heated at 105 °C for 15 min and dried at 80 °C for 48 h. The dried sample was mixed with the dry sample of tissue water content and passed through a 40-mesh sieve. Then, 0.03 g of the resulting dry sample was weighed for each replicate, and 6 mL of 80% ethanol was added. Samples were extracted in a water bath at 80 °C for 40 min to obtain an extract. Soluble sugar and reducing sugar were determined according to the method of Zhang et al. [13].

### 2.4. Data Analysis and Mapping

Four replicates were set up for each treatment, and their means and standard deviations were calculated. SPSS 13.0 for Windows (SPSS Inc., Chicago, IL, USA) was used to conduct independent sample t-tests. Data visualization was conducted with Origin 9.0 (OriginLab, Northampton, MA, USA).

## 3. Results

### 3.1. Soil Water Content at Different Drought Levels

The soil water content of the control group was relatively stable throughout the experimental cycle, remaining above 70%. Soil water content in the treatment group decreased as the number of drought days increased and was significantly different from the control group from day 9 onwards (Figure 1). On day 63, the soil water content of the treatment group was decreased by 93% compared with the control group.

### 3.2. Effect of Drought on Morphology

#### 3.2.1. Impact of Drought on External Morphological Characteristics

As shown in Figure 2, the morphological differences between the control and treatment groups gradually became obvious as the number of drought days increased. At the beginning of the drought (0–9 d), the control and treated plants were robust, with waxy glossy leaf surfaces and no obvious yellowing or wilting, and had high ornamental value. From day 18 to day 27, the drought-treated plants were still robust, with some leaves slightly drooping but with little change in morphology and leaf color. At the late stage of the drought (36–45 d), the leaves of the treated plants lost their luster and showed obvious yellowing and wilting, and the distance between leaves was continuously increased. At the end of the drought (54–63 d), the whole leaves of the treated plants were yellowed, and the plants lost their ornamental value (Figure 3A).

*Iris japonica* is a shallow-rooted plant with no obvious primary roots. After 63 days of drought, the rhizomes of both the control and treated groups appeared yellowish-brown (Figure 3B). The control group had a more developed fibrous root system that was concentrated near the rhizome, while the drought-treated fibrous roots were sparse and more elongated.

Table 1 presents the growth indicators of *I. japonica* after 63 days of drought. Compared with control plants, the leaf length, leaf mass ratio, and root mass ratio of plants under drought treatment were significantly decreased by 17%, 10%, and 22%, respectively, while the aboveground stem mass ratio and rhizome mass ratio of drought treatments significantly increased by 67% and 41%, respectively, compared with the control. The leaf width and root to shoot ratio were not significantly different between the control and treatment groups.

#### 3.2.2. Effects of Drought on Internal Leaf Anatomy

As shown in Figure 4, there was a difference in leaf anatomy between the drought treatment and the control after 63 d. The control and treated epidermis consisted of a layer of closely packed epidermal cells, with the control epidermal cells neatly arranged and the treated epidermal cells being relatively neat. There was no obvious differentiation between palisade tissue and sponge tissue in the leaves of *I. japonica*. At the same time, there was no regularity in the arrangement of mesophyll cells. The shape of the mesophyll cells in the control was mostly elliptical, while the drought-treated mesophyll cells were deformed to some extent.

Figure 5 shows the anatomical structure of the aboveground stem of *I. japonica*, and the epidermis, basic tissues (cortex and endodermis), and vascular bundles of the aboveground stem of *I. japonica* can be seen from the cross-sectional anatomical view. Both control and treatment plants had a tightly packed arrangement of epidermal cells. The lateral and medial basic tissues were separated by a thick layer of mechanical tissue composed of tightly packed, multi-layered, thick-walled cells. Except for epidermal cells, thick-walled mechanical tissue layers, and vascular bundles, control and treated cell sections were all elliptical or circular and closely aligned. In the basic tissues, the control and treatment contained a lot of collenchyma that were completely or partially dyed dark red or solid green. A large number of plasmolysis phenomena occurred in the drought-treated cells but not in the control cells.

Meanwhile, from both Table 2 and Figure 5, it can be seen that the number and size of vascular bundles in *I. japonica* stems on day 63 of drought treatment differed from that of the control. The vascular bundle diameter of *I. japonica* was reduced by 35% after drought treatment, which was a significant difference. However, the number of vascular bundles in the drought-treated group increased by 45% compared with the control. The cells of the basic tissue close to the epidermis were small, and the vascular bundles were scattered throughout the basic tissue, except the epidermis. Most of the vascular bundles were distributed deeper within the stems; the vascular bundles in the outer basic tissues were smaller, while the vascular bundles in the inner basic tissues were larger.

As can be seen in Figure 6, *I. japonica* is a monocotyledonous plant, and the roots only have a primary structure. In the anatomy of the roots, the control and treated epidermis consisted of a tightly packed epidermal cell. The cortex within the epidermis consisted of multiple layers of parenchyma cells in which the cells were loosely arranged with intercellular spaces. The parenchyma of the control cortex was larger with a loose arrangement, while the parenchyma cells that process the cortex were more closely arranged, without much of a gap between cells. Endothelial cells and vascular bundles were more pronounced in treated plants than in controls.

### 3.3. Effects of Water Content of Tissues under Different Drought Conditions

The water content of the leaves treated with drought gradually decreased as the number of drought days increased, and the difference was significant by 27 d. After 63 d, the drought treatment leaf water content was significantly lower than the control leaf water content, by 27% (Figure 7A). The water content of aboveground stems under drought conditions showed a slight increase and then decreased with the increase in drought days, beginning to decrease after 63 d, at which time it was significantly lower than the control, by 8% (Figure 7B). The water content of the rhizomes and roots in the drought treatment decreased as the number of drought days increased, and there were significant differences after 27 d and 9 d, respectively. After 63 d, the rhizome water content and root water content under the drought treatment were significantly lower than those of the control by 40% and 86%, respectively (Figure 7C,D).

### 3.4. Effect of Drought on Osmotic Adjustment Substances

As can be seen from Figure 8A,B, the content of proline and soluble protein in the drought-treated leaves significantly decreased after 9 d of drought and then significantly increased after 36 d and 27 d, respectively; compared with the control, it was decreased by 27% and 10%, respectively. After 63 d, the drought treatment leaf contents of proline and soluble protein significantly increased by 216% and 124%, respectively.

The content of soluble sugar in drought-treated leaves decreased as the number of drought days increased. The content of reducing sugar and soluble sugar in leaves showed a significant difference from 18 d to 27 d, respectively. After 63 d of drought, the contents of leaf reducing sugar, leaf soluble sugar, and aboveground stem soluble sugar in the treated leaves decreased by 72%, 51%, and 55%, respectively (Figure 8C–E). However, the difference in root soluble sugar content under drought treatment was not significant.

### 3.5. Effect of Drought on Photosynthetic Pigments

As Table 3 shows, compared with the control, the content of photosynthetic pigments was significantly reduced under the drought stress treatment, with chlorophyll a content, chlorophyll b content, and chlorophyll (a + b) content reduced by 37%, 32%, and 36%, respectively, after 63 d of drought, compared with the control. The drought treatment reduced chlorophyll a content, chlorophyll b content, and chlorophyll (a + b) content by 37%, 32%, and 36%, respectively, compared with the control. Carotenoid content was most significantly reduced under stress compared with the control, being reduced by 41%. However, the chlorophyll a/b content of the drought treatment was only 7% lower than that of the control.

### 3.6. Effect of Drought on Oxidation System and Membrane Lipid Peroxidation Products

As shown in Figure 9A, the AsA content of drought-treated leaves decreased as the drought days increased, and there was a significant difference from the control after 18 d, which was significantly lower than the control by 64% after 63 d; the GSH content of the drought-treated leaves first increased and then decreased as the number of drought days increased (Figure 9B); it increased by 28% after 9 d compared with the control and significantly decreased by 2% after 63 d.

As shown in Table 4, the activity levels of APX, GR, and MDHAR in the drought-treated leaves decreased after 63 d of drought, while the activity of AAO increased; the activities of the four enzymes showed significant differences. The activities of APX, GR, and MDHAR in drought-treated leaves decreased by 29%, 40%, and 22%, respectively, and the activity of AAO increased by 169% compared with the control. As the number of days increased, the activity levels of leaf POD and SOD increased significantly, by 432% and 91%, respectively, after 63 d.

As can be seen from Figure 9C, the cell membrane permeability of the drought-treated leaves increased with the number of drought days, and a significant difference began to appear at 27 d. After 63 d, cell membrane permeability increased by 240% compared with the control; the aldehyde content increased significantly by 44%, respectively, after 63 d compared with the control (Table 4). On day 63, there was a significant difference in hydrogen peroxide content between the drought treatment and the control, with a 145% increase in the former relative to the control (Table 4).

### 3.7. Effect of Drought on Root Activity

As shown in Figure 10, root vigor under the drought treatment showed a continuous decline as the number of days of drought increased. The decline was more moderate from 0 to 9 d, with a more significant difference starting after 18 d. The most pronounced decline in root vigor was observed after 18–27 d. From 27 d onwards, root vigor continued to slowly decline, then increased at 45 d, and continued to slowly decline from 45 d to 63 d. After 63 d, the drought treatment’s root vigor was already 77% lower than the control’s root vigor.

## 4. Discussion

### 4.1. Morphological and Anatomical Characteristics of Plants under Drought Conditions

Plants subjected to drought stress first respond through changes in external morphology, including plant height, leaf size, and degree of root development [21]. Chen et al. [22] found that the root length, number of lateral roots, and root to shoot ratio of wheat increased as the number of days of drought stress increased, indicating plant regulate root growth to adapt to drought. In the present study, after 20 d of drought treatment, the leaf length of *I. japonica* became shorter, and the ratio of leaf weight to root weight decreased; however, the ratio of branch weight to root and rhizome increased. This indicated that *I. japonica* adapted to drought by increasing its proportion of stem, similar to the results of previous research [23]. This may be related to the growth habit of *I. japonica* tending to form new clones through rhizome cloning and reproduction, which occurs in its natural environment when *I. japonica* continuously elongates the apex of its rhizome and grows into new asexual plantlets. Therefore, when *I. japonica* is subjected to drought stress, it mitigates the stress by regulating various physiological changes in the stem.

In addition, changes in various cellular tissues of *I. japonica* were also studied. Previous studies showed that drought leads to a loss of membrane fluidity and induces structural and morphological changes in the cell wall [24,25]. This was also verified in our study; *I. japonica* leaves had no fissures and no spongy tissue but instead had well-developed aeration tissue. Anatomically, the drought-treated leaf cells were more closely arranged after 63 d and showed extensive cell lysis, indicating that *I. japonica* had taken measures to reduce water evaporation, but the leaf cells were unable to withstand intense long-term drought stress, as can be seen in Figure 3, by the end of which the rhizomes of *I. japonica* were too severely desiccated to be sectioned. Additionally, probably because of the reduced water content of *I. japonica* leaves, the plants were dehydrated, making it impossible to maintain the cells in a normal state, resulting in the mesophyll cells being irregular in shape; whereas, in terms of the root anatomy, the phenotype of the thin-walled cell gaps was large and loose, probably enabling the maintenance of good ventilation under adequate water [23,26]. Vascular bundles are the main tissues for water transport in plant stems [27]. Owing to the loss of cellular water, the diameter of vascular bundles under drought was significantly reduced, which, in turn, led to a decrease in water transport efficiency. The significant increase in the number of vascular bundles may be a response of *I. japonica* to drought stress, i.e., a “numerical advantage” to partially offset the reduced efficiency of vascular water transport.

### 4.2. Response of Tissue Water Content to Drought

The water content of tissues directly reflects the water content of the plant parts. Under drought stress, the reduction in water content in plants can differently respond, depending on the characteristics of the different parts of the plant [28]. Luo et al. [29] studied *Medicago sativa* L. seedlings and found that their roots were more resistant to drought stress than their leaves or stems. *Iris japonica* has different growth characteristics from lucerne seedlings, and, as a perennial flower, the root and stem of *I. japonica* play an important role in its growth [30,31]. The present results showed that the water content of the leaves, rhizomes, and roots of *I. japonica* significantly decreased after 27 d, but the aboveground stems did not show severe drought stress until day 63. This suggested that under drought stress, *I. japonica* leaves withered first, protecting the stem part so that new leaves would grow after moisture might be restored. At the same time, it could be the cells of the aboveground stem being closely arranged and containing a thick-walled mechanical tissue layer that reduced the loss of water [32].

### 4.3. Response of Osmotic Adjustment Substances to Drought

Osmoregulatory substances are often used as important indicators of drought tolerance in plants. For example, *Lagerstroemia indica*, *Glycine max*, and *Arabidopsis thaliana* regulate their soluble sugars, proline, and other substances to alter the content of osmoregulatory substances, thereby increasing their drought tolerance [33]. In our study, the response of *I. japonica* was not obvious in the early stage of drought, but, with the continuation of drought, the proline and soluble protein content of *I. japonica* gradually increased, which increased the osmotic pressure of protoplasts, thereby preventing water dissipation, thus acting as a water-holding agent for protoplasts as a way to better cope with drought stress. The results are consistent with those observed in *Lycium ruthenicum* [34].

Crops, such as *Oryza sativa* and *Triticum aestivum*, produce certain small molecule metabolites, which include substances such as soluble sugars, in response to drought stress to counteract their own drought stress [35,36]. Soluble sugars are not only an osmoregulatory substance but also a source of energy for plant growth and development. Soluble sugars include reducing sugars and non-reducing sugars, of which reducing sugars are one of the discriminators of drought adaptation in plants [37]. The contents of leaf reducing sugar, leaf soluble sugar, and aboveground stem soluble sugar under drought treatment of *I. japonica* decreased as the number of drought days increased, indicating that the material accumulation of leaves and aboveground stems of *I. japonica* decreased, which may be the reason for the weakening of photosynthesis in *I. japonica* and the concomitant decrease in sugars in photosynthetic products caused by drought. Our results are similar to those observed in *Sesuvium portulacastrum* [38].

### 4.4. Effect of Drought Stress on Plant Biochemical Indicators

Chlorophyll and carotenoids are important photosynthetic pigments. When exposed to drought stress, the chloroplast lamellar structure is damaged, and chlorophyll and carotenoid synthesis is impaired, which, in turn, affects electron transport and photosynthetic phosphorylation processes [39,40]. After being subjected to drought stress for 63 days, all photosynthetic pigment indicators of *I. japonica* were reduced, which is consistent with the results of previous studies [41,42]. The reduction of these indicators implies that the photosynthetic capacity of *I. japonica* was weakened, which may have reduced the content of sugar (acting as an osmoregulatory substance), thus affecting the growth and development of *I. japonica*.

Cell membrane permeability, hydrogen peroxide concentration, and MDA content can reflect the degree of damage to cell membranes. Hydrogen peroxide is a reactive oxygen species that causes strong oxidative damage to cell membrane lipids during plant metabolism [43]. MDA is the end product of lipid peroxidation in plant membranes, and its content reflects the degree of damage in plants. Drought increased the cell membrane permeability and the content of hydrogen peroxide and MDA in *I. japonica*, which is consistent with previous research results [44].

Plants evolved the ability to scavenge ROS through an enzymatic scavenging system consisting of antioxidant enzymes, such as SOD and POD, over a long evolutionary period, which, in turn, inhibits membrane lipid peroxidation and enhances plant drought resistance [45]. Previously, the activities of antioxidant substances (AsA and GSH) and antioxidant enzymes (APX, MDHAR, and GR) in the saline-resistant glutathione cycle and the resistance to enzymatic reactions were investigated to elucidate the process of plant responses to drought stress [46,47].

AsA has the ability to directly scavenge various reactive oxygen species generated by stress and plays an important role in plant stress resistance [48]. The oxidative breakdown of AsA in plant cells is mainly catalyzed by AAO and APX, and MDHAR is one of the enzymes involved in AsA regeneration. It was shown that an increase in AAO activity adversely affects photosynthesis [49]. In the present study, the increase in AAO activity in the drought treatment accelerated the rate of AsA catabolism, which exacerbated the damage caused by drought in *I. japonica*. Meanwhile, the activity of MDHAR was reduced under drought, which was detrimental to AsA production. Finally, the AsA content of *I. japonica* decreased. This series of changes indicated that *I. japonica* was unable to resist the damage caused by 63 days of drought. In contrast, GSH is also an antioxidant that can effectively scavenge reactive oxygen species, and GR is one of the enzymes involved in GSH regeneration. After drought treatment, GSH content decreased with time, indicating that mild drought could indeed promote the antioxidant capacity of *I. japonica*, but, as the drought continued, GSH content and GR activity decreased, stress exceeded the tolerance threshold of plants, membrane lipid peroxidation increased, and normal cell metabolism was disturbed [50]. POD is the main enzyme involved in scavenging hydrogen peroxide in plants, while APX and SOD play important roles in scavenging superoxide anion radicals [51]. A decrease in APX activity would affect AsA metabolism [52]. POD and SOD activities increased under drought, which indicates that *I. japonica* has a strong ability to reduce ROS accumulation to maintain membrane system stability and enhance plant drought resistance [53].

Based on the above findings, the ascorbate–glutathione cycle of *I. japonica* was affected by the accumulation of harmful substances under drought conditions, mainly through the increase in POD and SOD activities to resist the cellular damage caused by ROS.

### 4.5. Dynamic Changes in Root Vigor

Root activity directly reflects plant growth. Chen et al. found that when *T. aestivum* was subjected to drought stress, not only did the root length and root dry weight significantly decrease but also the root activity was also affected to some extent [22]. In the present study, the results of the root morphology of *I. japonica* were similar to those of the abovementioned studies. The water content and root activity of *I. japonica* roots began to decrease at 9 d and 18 d, respectively, indicating that the response of *I. japonica* roots to water supply was sensitive.

## 5. Conclusions

In this study, *I. japonica* mainly physiologically adapted to drought by increasing the proportion of stems in the plant to tolerate drought stress. The increasing proline and soluble protein content of the leaves and contents of POD and SOD played important roles in allowing *I. japonica* to relieve drought conditions. These findings indicated the mechanism by which *I*. *japonica* is able to keep growing well under prolonged droughts. Given its excellent physiological drought response, *I. japonica* holds promise as an ornamental plant to be grown in urban areas that are subject to drought. While also saving municipal water, the planting of *I*. *japonica* also facilitates simplified management and reduces the maintenance cost, which can not only improve the green coverage of a city but also facilitate the construction of ecological gardens.

## Figures and Tables

**Figure 1 plants-12-03729-f001:**
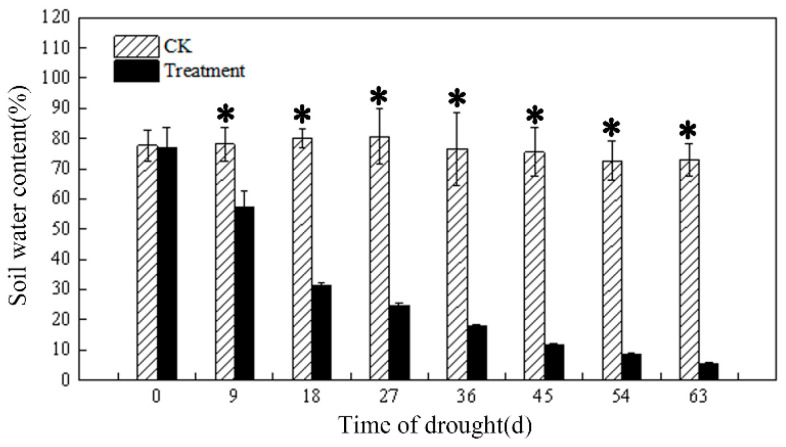
Changes in soil water content across drought days. Notes: the values are represented as means ± standard error (*n* = 4). The * in the figure indicates significant differences between control and drought treatments (*p* < 0.05).

**Figure 2 plants-12-03729-f002:**
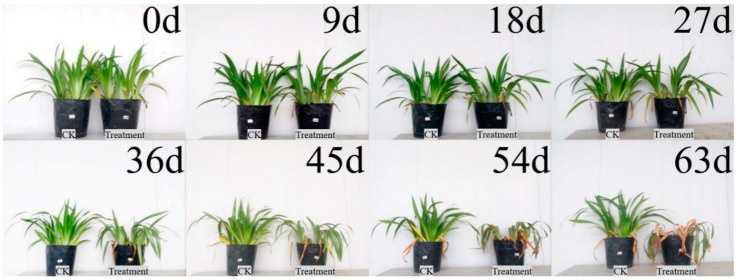
Effects of different drought-treatment times on *Iris japonica* growth status.

**Figure 3 plants-12-03729-f003:**
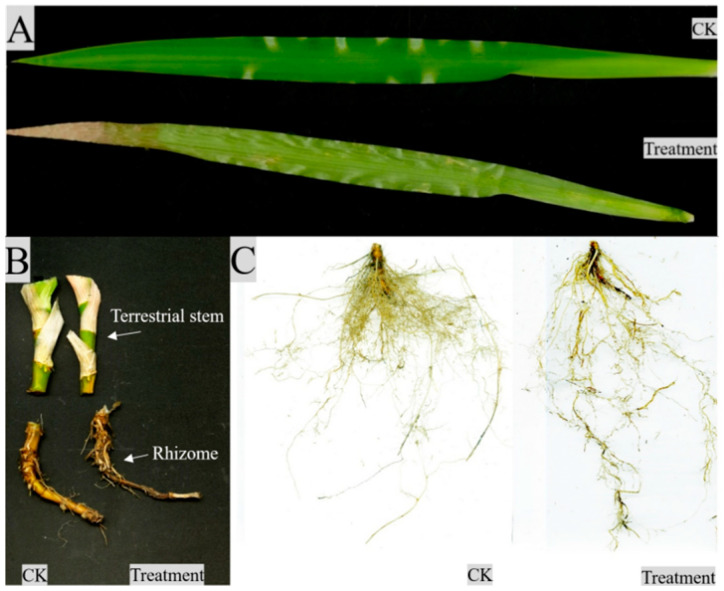
Changes in leaves, terrestrial stems, rhizomes, and roots of *Iris japonica* during drought on day 63. (**A**) Leaf changes of *I. japonica* after 63 days of treatment; (**B**) terrestrial stem and rhizome changes of *I. japonica* after 63 days of treatment; (**C**) root changes of *I. japonica* after 63 days of treatment.

**Figure 4 plants-12-03729-f004:**
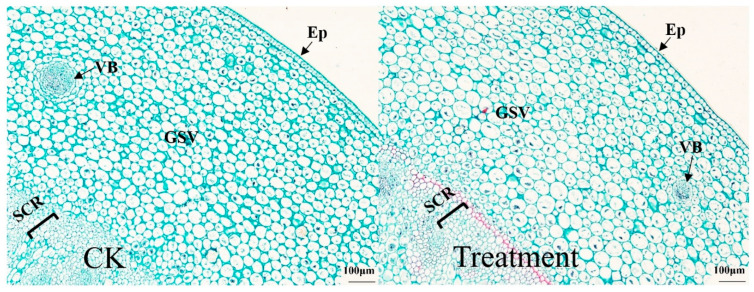
Effects of drought on leaf anatomy on day 63. CK, control; Ep, epidermal cells; Me, mesophyll cells; VB, vascular bundle.

**Figure 5 plants-12-03729-f005:**
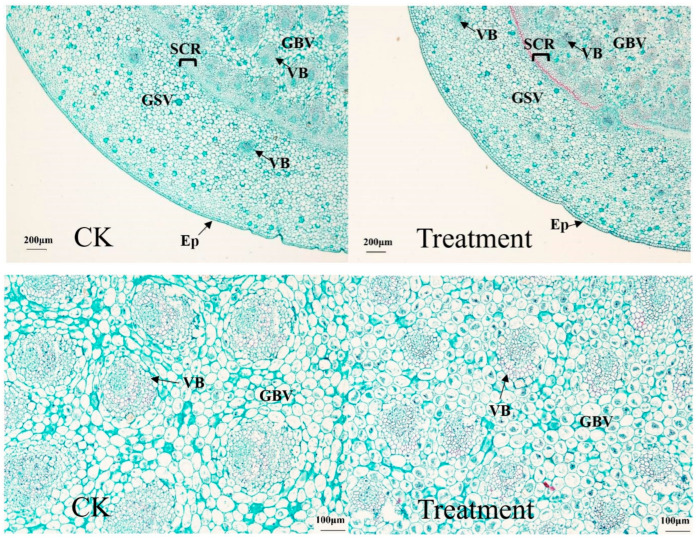
Effects of drought on terrestrial stem anatomy on day 63. CK, control; Ep, epidermal cells; VB, vascular bundle; GBV, ground tissue with big vascular bundle; GSV, ground tissue with small vascular bundle; SCR, sclerenchyma ring.

**Figure 6 plants-12-03729-f006:**
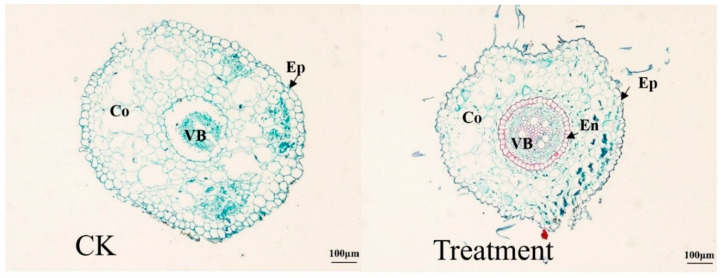
Effects of drought on root anatomy on day 63. CK, control; Ep, epidermal cells; Co, cortex; En, endodermis; VB, vascular bundle.

**Figure 7 plants-12-03729-f007:**
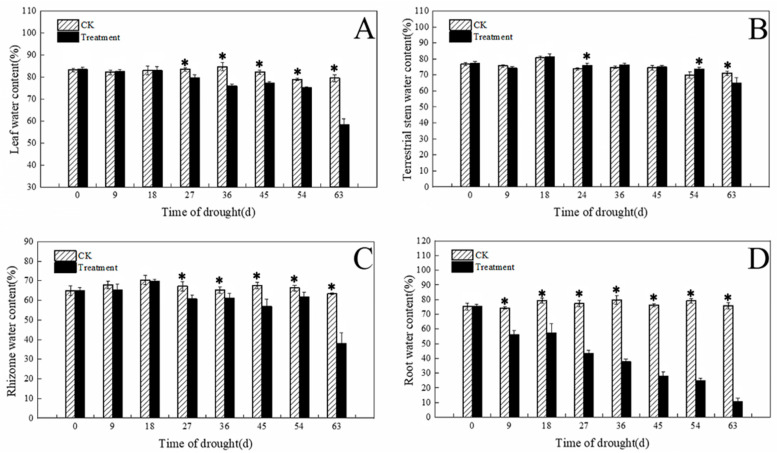
Water content of tissues under different drought conditions. (**A**) Leaf water content; (**B**) terrestrial stem water content; (**C**) rhizome water content; (**D**) root water content. The * in the figure indicates significant differences between control and drought treatments (*p* < 0.05).

**Figure 8 plants-12-03729-f008:**
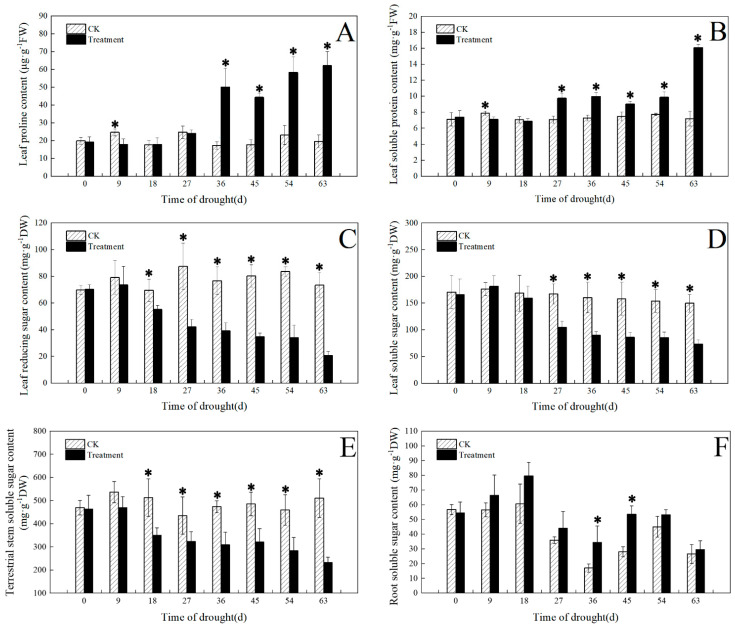
Effects of drought on osmotic regulatory substances. (**A**) Leaf proline content; (**B**) leaf soluble sugar content; (**C**) leaf reducing sugar content; (**D**) leaf soluble sugar content; (**E**) terrestrial stem soluble sugar content; (**F**) root soluble sugar. The * in the figure indicates significant differences between control and drought treatments (*p* < 0.05).

**Figure 9 plants-12-03729-f009:**
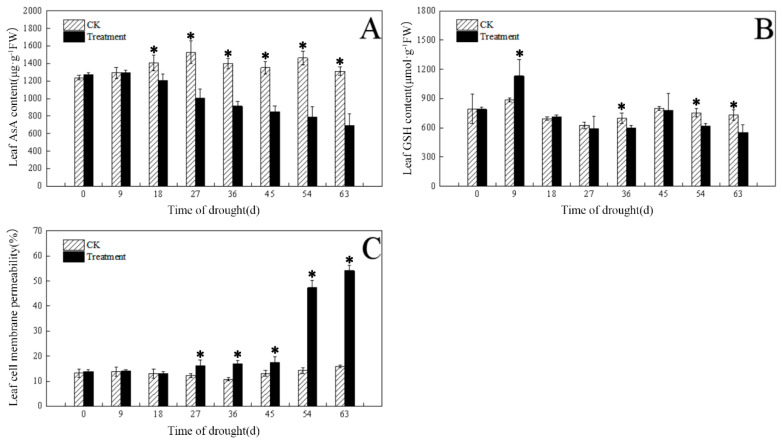
Effect of drought on ascorbic acid (AsA), glutathione (GSH), and cell membrane permeability in leaves. (**A**) Leaf AsA content; (**B**) leaf GSH content; (**C**) leaf cell membrane permeability. The * in the figure indicates significant differences between control and drought treatments (*p* < 0.05).

**Figure 10 plants-12-03729-f010:**
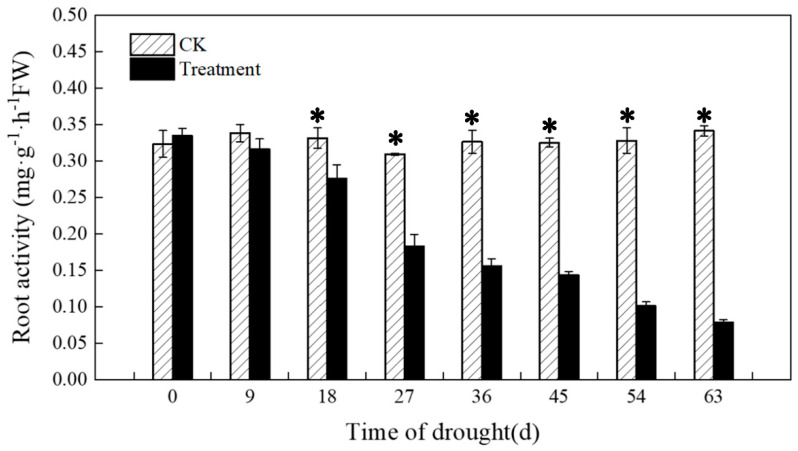
Effects of drought on root activity. The * in the figure indicates significant differences between control and drought treatments (*p* < 0.05).

**Table 1 plants-12-03729-t001:** Changes in growth indices of *Iris japonica* on day 63 of drought. Notes: data are means ± SD. Different lowercase letters denote significant differences between treatments (*p* < 0.05).

Growth Indices	Control (ck)	Treatment
Leaf length (cm)	38.70 ± 1.09 a	32.13 ± 0.85 b
Leaf width (cm)	2.53 ± 0.13 a	2.30 ± 0.14 a
Leaf mass ratio	0.73 ± 0.01 a	0.66 ± 0.05 b
Aboveground stem mass ratio	0.11 ± 0.01 b	0.18 ± 0.02 a
Rhizome mass ratio	0.06 ± 0.01 b	0.08 ± 0.02 a
Root mass ratio	0.10 ± 0.01 a	0.08 ± 0.01 b
Root to shoot ratio	0.19 ± 0.01 a	0.19 ± 0.04 a

**Table 2 plants-12-03729-t002:** Comparison of vascular tissue structure in the cross section of aboveground stems of *Iris japonica* on day 63 of drought. Different lowercase letters denote significant differences between treatments (*p* < 0.05).

Organizational Structure of Vascular Bundles of Aboveground Stems in Cross-Section	Control (CK)	Treatment
Number of vascular bundles	9.67 ± 1.15 b	14.00 ± 1.00 a
Vascular bundle diameter (μm)	221.32 ± 45.56 a	144.78 ± 25.30 b

**Table 3 plants-12-03729-t003:** Content of photosynthetic pigments on drought day 63. Different lowercase letters denote significant differences between treatments (*p* < 0.05).

Photosynthetic Pigment	Control (ck)	Treatment
Chlorophyll a content (mg·g^−1^ fresh weight (FW))	0.67 ± 0.08 a	0.42 ± 0.03 b
Chlorophyll b content (mg·g^−1^ FW)	0.28 ± 0.02 a	0.19 ± 0.01 b
Carotenoid content (mg·g^−1^ FW)	0.17 ± 0.03 a	0.10 ± 0.01 b
Chlorophyll (a + b) content (mg·g^−1^ FW)	0.95 ± 0.10 a	0.61 ± 0.04 b
Chlorophyll a/b	2.33 ± 0.11 a	2.17 ± 0.02 b

**Table 4 plants-12-03729-t004:** Peroxidase (POD) activity, superoxide dismutase (SOD) activity, ascorbate peroxidase (APX) activity, glutathione reductase (GR) activity, monodehydroascorbate reductase (MDHAR) activity, ascorbic acid oxidase (AAO) activity, malondialdehyde (MDA) activity and hydrogen peroxide content in leaves during drought day 63. Different lowercase letters denote significant differences between treatments (*p* < 0.05).

Enzyme Activity Indices	Control (ck)	Treatment
Leaf POD activity (U·g^−1^ min^−1^ fresh weight (FW))	24.31 ± 1.87 b	129.45 ± 4.86 a
Leaf SOD activity (U·g^−1^ h^−1^ FW)	409.74 ± 67.61 b	783.11 ± 17.27 a
Leaf APX activity (U·g^−1^ min^−1^ FW)	53.22 ± 1.84 a	37.84 ± 3.86 b
Leaf GR activity (U·g^−1^ min^−1^ FW)	12.95 ± 1.34 a	7.82 ± 1.01 b
Leaf MDHAR activity (U·g^−1^ min^−1^ FW)	20.71 ± 1.70 a	16.18 ± 0.93 b
Leaf AAO activity (U·g^−1^ min^−1^ FW)	4.27 ± 0.70 b	11.49 ± 1.34 a
Leaf MDA activity (mmol·g^−1^ FW)	6.29 ± 0.54 b	9.07 ± 0.40 a
Leaf hydrogen peroxide content (mmol·g^−1^ FW)	1413.16 ± 51.53 b	3464.13 ± 131.64 a

## Data Availability

Not applicable.

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
