# Peer review of "Morphological Structure and Physiological and Biochemical Responses to Drought Stress of Iris japonica"

_plants, 2023, doi:10.3390/plants12213729_

Round 1
Reviewer 1 Report
Comments and Suggestions for Authors
First of all I appreciate your work. However it can be improved after minor revision.
Line 11 and 13: Please write in italic font Iris japonica.
Line 17-18: Please write percentage increase of all parameters in your study
Line 20. Please write in italic font and carefully checked in whole manuscript.
Line 23: There are too much key words. Please delete that one which are already used in the title.
Line 27: In China, nearly half of the region is exposed to severe drought. How much? any percentage? Please make it clear.
What is significance of your study on Iris japonica?
Vertical title is bold in some graphs while un bold in others. Please make sure it should be in consistent way.
Please improve conclusion part also.
Comments on the Quality of English LanguageThere is need to improve English.
Reviewer 2 Report
Comments and Suggestions for Authors
The manuscript entitled “Morphological structure and physiological and biochemical responses of Iris japonica to drought stress” is devoted to an interesting topic. This research produced some valuable results. However, due to different sampling times, it is not possible to draw a conclusion on drought tolerance. I believe that only results observed after sampling and measuring all morpho-anatomical, physiological, and biochemical changes at the same time during the experimental period can give information on plant status in given growth conditions.
In my opinion, this manuscript needs to be improved in all sections. The introduction section is not informative enough. In the material and methods, most of the methods and experimental design are not clearly described. Results should also be clearly written. Figure and table captions should be rewritten too, in their present form they are rather confusing and partly incorrect. The results should be statistically processed so that treatments can be compared with each other.
I hope some of the comments inserted into the pdf, and written with good intentions, will help improve the experiment and manuscript. Some of the phrases and sentences that are not well structured are underlined with a green line.
Best regards

Comments on the Quality of English Language-
Reviewer 3 Report
Comments and Suggestions for Authors
The paper by Yu et al describes the structural and physiological changes that occur in Iris japonica in response to drought stress. The paper presents a thorough analysis in order to support findings that indicate changes at a structural and physiological level. There are however certain points that I would like to stress:
1. Summary; please restructure the summary e.g the sentence "I. japonica can regulate....... soluble proteins" should not be there. Also there are mistakes e.g . "plant plants". Also use commas for separating phrases and not "and". The summary does not highlight any findings that support the conclusion for the promotion of I. Japonica in water-scarce areas.
2. Introduction: the authors should check the meaning of the sentences they use e.g "abiotic stresses represented by drought" is not subjective.
3. Results : The authors use a range of techniques to assess drought effect including morphological study, measure of water content, proline content, cell membrane permeability, soluble protein, hydrogen peroxide and antioxidants, AAO activity, photosynthetic pigments, root vigour and sugars.
There is however, no statistical analysis to support their morphological data and images provided are difficult to interprit. I.e. authors could use a number of microscopy profiles and apply a stereological method that would convert their observations to measurable values on which they could perform statistical analysis.
In addition a molecular marker (or markers) could be also used to follow the drought effect on plant stages used for the other measures taken.
Concluding, the paper presents an interesting study providing data on important measurable parameters that could be taken into account in order to study drought stress. It would be however useful if they could provide more concrete data supported by a basic molecular study.
Comments on the Quality of English LanguagePlease check language with a professional English speaking.
Round 2
Reviewer 2 Report
Comments and Suggestions for Authors
Dear Authors,
Thank you for considering all comments and suggestions. After the corrections were made, the manuscript was significantly improved.
Best regards